# Circulating Chromosome Conformation Signatures Significantly Enhance PSA Positive Predicting Value and Overall Accuracy for Prostate Cancer Detection

**DOI:** 10.3390/cancers15030821

**Published:** 2023-01-29

**Authors:** Dmitri Pchejetski, Ewan Hunter, Mehrnoush Dezfouli, Matthew Salter, Ryan Powell, Jayne Green, Tarun Naithani, Christina Koutsothanasi, Heba Alshaker, Jiten Jaipuria, Martin J. Connor, David Eldred-Evans, Francesca Fiorentino, Hashim Ahmed, Alexandre Akoulitchev, Mathias Winkler

**Affiliations:** 1School of Medicine, University of East Anglia, Norwich NR4 7TJ, UK; 2Oxford BioDynamics Limited, Oxford OX4 2WB, UK; 3Department of Surgery and Cancer, Imperial College London, London SW7 2AZ, UK; 4Department of Biotechnology, Amity University, Noida 201313, India; 5Nightingale-Saunders Clinical Trials and Epidemiology Unit, King’s College London, London WC2R 2LS, UK

**Keywords:** prostate cancer, diagnosis, screening, epigenetics, nucleome, PSA, blood test

## Abstract

**Simple Summary:**

Prostate cancer occurs in one out of six men during their lifetime. Because its symptoms are not specific, it is often diagnosed late. The widely used prostate-specific antigen (PSA) blood test does not have sufficient accuracy, resulting in numerous unnecessary prostate biopsies in men with benign disease and false reassurance in some men with cancer. We have recently developed an epigenetic test for prostate cancer that detects cancer-specific chromosome conformations in the blood of the patient. In this study, we combined this epigenetic test with the PSA test and used two cohorts of patients to determine whether they have better diagnostic accuracy when used together. Our results demonstrate that the new combined test (termed PSE test) allows significant increase in prostate cancer detection compared to PSA or epigenetic test alone. This new PSE test is accurate, rapid, minimally invasive, and inexpensive. If successful in larger trials, it may significantly improve prostate cancer diagnosis.

**Abstract:**

Background: Prostate cancer (PCa) has a high lifetime prevalence (one out of six men), but currently there is no widely accepted screening programme. Widely used prostate specific antigen (PSA) test at cut-off of 3.0 ng/mL does not have sufficient accuracy for detection of any prostate cancer, resulting in numerous unnecessary prostate biopsies in men with benign disease and false reassurance in some men with PCa. We have recently identified circulating chromosome conformation signatures (CCSs, Episwitch^®^ PCa test) allowing PCa detection and risk stratification in line with standards of clinical PCa staging. The purpose of this study was to determine whether combining the Episwitch PCa test with the PSA test will increase its diagnostic accuracy. Methods: *n* = 109 whole blood samples of men enrolled in the PROSTAGRAM screening pilot study and *n* = 38 samples of patients with established PCa diagnosis and cancer-negative controls from Imperial College NHS Trust were used. Samples were tested for PSA, and the presence of CCSs in the loci encoding for of *DAPK1*, *HSD3B2*, *SRD5A3*, *MMP1*, and *miRNA98* associated with high-risk PCa identified in our previous work. Results: PSA > 3 ng/mL alone showed a low positive predicted value (PPV) of 0.14 and a high negative predicted value (NPV) of 0.93. EpiSwitch alone showed a PPV of 0.91 and a NPV of 0.32. Combining PSA and Episwitch tests has significantly increased the PPV to 0.81 although reducing the NPV to 0.78. Furthermore, integrating PSA, as a continuous variable (rather than a dichotomised 3 ng/mL cut-off), with EpiSwitch in a new multivariant stratification model, Prostate Screening EpiSwitch (PSE) test, has yielded a remarkable combined PPV of 0.92 and NPV of 0.94 when tested on the independent prospective cohort. Conclusions: Our results demonstrate that combining the standard PSA readout with circulating chromosome conformations (PSE test) allows for significantly enhanced PSA PPV and overall accuracy for PCa detection. The PSE test is accurate, rapid, minimally invasive, and inexpensive, suggesting significant screening diagnostic potential to minimise unnecessary referrals for expensive and invasive MRI and/or biopsy testing. Further extended prospective blinded validation of the new combined signature in a screening cohort with low cancer prevalence would be the recommended step for PSE adoption in PCa screening.

## 1. Background

PCa (PCa) is the most commonly diagnosed internal cancer in men and is the second leading cause of cancer-related death in the Western world [1,2]. It is common and most men aged above 80 will show some evidence of microscopic PCa [3,4]. Many of these cancers may not manifest clinically. Unfortunately, some cases of PCa represent aggressive disease, which significantly reduces patients’ life expectancy. It is often difficult to distinguish such cases, especially at an early stage. PCa is diagnosed using a prostate-specific antigen (PSA) blood test and magnetic resonance imaging (MRI), followed by targeted needle biopsy [5,6]. 

Clinically widely used PSA blood test is not very accurate for PCa diagnosis. Currently 3.0 ng/mL is used as an indicator for further investigation. At this cut-off the test has a sensitivity of 59 percent with specificity of 87 percent for detection of any PCa (including insignificant slow growing PCa), resulting in many unnecessary prostate biopsies in men with benign disease [7]. Alternatively, men with PSA lower than 3.0 ng/mL may also have cancer, some of which may be aggressive [8]. In early PCa, PSA testing cannot differentiate between early-stage invasive cancers and latent, non-lethal tumours that would have remained asymptomatic during a man’s lifetime. Several new blood tests have emerged for PCa detection including 4K blood test (AUC 0.8), Stockholm 3 score (AUC 0.75) and PHI blood test (90% sensitivity, 17% specificity) [9]. 

PCa screening could be beneficial if the screening tests are affordable and more accurate. A large European Randomized study of screening for PCa (ERSPC) that included 182,160 men, followed up to 16 years showed the rate ratio of PCa mortality of 0.80 (*p* < 0.001). To prevent one PCa death, the number needed to screen was 570, and the number needed to diagnose was 18 [10]. The main clinical problem of the currently available blood tests for PCa remains their low positive predictive value (PPV) which results in unnecessary referrals to secondary care and expensive and invasive testing like MRI and biopsies. 

In the recently published PROSTAGRAM pilot study, we have compared the screening value of PSA, ultrasound (USS), and MRI in detecting PCa in a self-referred population of men older than 50 years of age using a combination of invitation and advertisement. All three modalities had similar accuracies of ~0.8 with PSA and MRI having slight advantage over USS (Table 1) [11]. Interestingly each screening modality had high NPV values of 0.93–0.95 and low PPV values of 0.23–0.32, which is similar to previous studies [12]. For patients, this leads to high rate of referral for biopsies which often yield no cancer or insignificant disease [11].

With the advent of epigenetic research, it has become evident that epigenetic modifications like aberrant DNA methylation [13,14] and histone acetylation [15,16] are related to PCa onset. Three dimensional chromatin conformations (CCs), as part of genomic regulatory architecture, are also powerful epigenetic regulators of gene expression and cellular pathological phenotypes [17]. Long-range epigenetic alterations in CCs were found in primary prostate tumours and circulating DNA from PCa patients [18]. We have previously developed an epigenetic assay EpiSwitch [19] that employs an algorithmic-based CCs analysis (Figure 1). Using EpiSwitch technology, we have shown the presence of cancer-specific CCs in peripheral blood mononuclear cells (PBMCs) and primary tumours of patients with melanoma [20,21] and PCa [22]. In light of the regulatory role attributed lately in pathology to systemic exosome traffic, we have now used indirect co-culture experiment transfer and demonstrated horizontal transfer of CCs between cultured PCa cells and monocytes without direct contact [23]. 

In this study we have used available full blood samples from patients enrolled into a prospective PCa screening pilot study (PROSTAGRAM) [11] (*n* = 109) as well as *n* = 38 samples of patients with either an established PCa diagnosis or confirmed cancer-negative controls from Imperial College NHS Trust (Table 2) to determine whether combining Episwitch PCa test with PSA test will increase its diagnostic accuracy.

## 2. Methods

### 2.1. Patient Population

In this retrospective case-control study, *n* = 109 whole blood samples (*n* = 88 controls and *n* = 21 cancers) of men 50–69 years of age enrolled in the PROSTAGRAM screening study (described in [11]), and *n* = 38 samples of patients with either established PCa diagnosis *n* = 29 or cancer-negative controls *n* = 9 from Imperial College NHS Trust were used (described in [22]) (Table 2). Samples were randomly allocated for training and test cohorts with sample ratio ~1:2. The study was approved by the UK National Research Ethics Committee and conducted in accordance with Good Clinical Practice guidelines and the Declaration of Helsinki. All participants provided written informed consent. All data were pseudo-anonymized. All procedures and protocols were performed in accordance with the relevant guidelines and regulations. 

### 2.2. Sample Preparation

A 5 mL blood sample was collected from PCa patients and controls using BD Vacutainer^®^ plastic EDTA tubes. The tubes were passively frozen and stored at −80 °C. DNA from the whole cell lysate was isolated and fixed with formaldehyde (Figure 1). To study interchromatin associations, fixed chromatin was digested into fragments with TaqI restriction enzyme, and the DNA strands were joined favouring cross-linked fragments. The cross-links were reversed, and PCR performed using the primers previously established by the EpiSwitch software (as previously described in detail in [20,21,22,24]). 

### 2.3. EpiSwitch PCR

EpiSwitch technology platform (Oxford Biodynamics, Oxford, UK) pairs high resolution 3C results with regression analysis and a machine learning algorithm to develop disease classifications [20,21,24,25]. Samples were tested for PSA and the presence of CCSs in the loci of *DAPK1*, *HSD3B2*, *SRD5A3*, *MMP1*, and *miRNA98* associated with high risk PCa identified in our previous work (EpiSwitch PCa test) [22]. The exact PCR protocol was described previously [20,21,22,24]. 

### 2.4. Statistical Analysis

All analysis for this study was performed using libraries which are developed for the R Statistical Language (R version 4.1.2). Feature engineering of the EpiSwitch Markers, with either binary PSA or continuous PSA was performed using Recursive Feature Elimination (RFE) using LDA, Xgbtree, xgblinear, decision trees, and random forest libraries. The XGBoost algorithm model described in [26] was used for final test optimisation. The grid search algorithm was used to optimize the hyper-parameters and learning rate in each iteration. For drawing inferences, we used SHapley Additive exPlanations (SHAP) values that are computed by a game theoretical approach which quantifies the contribution of each feature within a model to the final prediction of an observation [27]. To control for batch effects (non-biological variation), due to the measurement technology qPCR, and to allow for long term use of the final produced model(s), we have implemented a Batch adjustment by reference alignment (BARA) procedure [28] (Figure 2) on the results of PSA and EpiSwitch datasets.

## 3. Results

### 3.1. Bayesian Test Performance for Individual and Combined Cohorts

In the isolated PROSTAGRAM cohort of 109 men, PSA at cut-off of 3 ng/mL had a PPV of 0.14 (95% CI 4.3% to 77.7%) and a NPV of 0.93 (95% CI 89.6% to 96.4%) with accuracy of 0.79 (95% CI 69.2% to 88.0%) (Table 3). When used as a continuous variable, PSA has lower PPV of 0.08 (95% CI 2.8% to 15.8%) and a NPV of 0.93 (95% CI 82.4% to 98.0%) (Table 3). EpiSwitch PCa test alone has a PPV of 0.91 (95% CI 83.16% to 96.0%) and a NPV of 0.32 (95% CI 24.8% to 40.2%) with combined accuracy of 0.64 (95% CI 54.5% to 73.2%) (Table 3).

### 3.2. Test Optimization

We have then used the Recursive Feature Elimination, XGBoost algorithm model and BARA procedure as described in statistical section for final Episwitch-PSA test optimisation. In the process of test optimisation, the Classification and Regression Training (CARET) package with recursive feature elimination (RFE) was applied using a random forest classifier. This approach takes in 1:*n* different number of markers with random selection and builds different models with random combinations. the *ERG_21* marker (previously published in [22] has shown to have low significance in disease prediction and was removed (see further details in the Materials and Methods Statistical Analysis section).

### 3.3. Evaluation of Test Performance in the Combined Cohort Comparing Continuous and Dichotomous PSA

We have first analysed the datasets with a cut-off of PSA at 3 ng/mL (as in the PROSTAGRAM study [11]). Samples were randomly allocated and optimally balanced. Tests were trained on 51 samples (23 non-cancer, 28 cancer) and then tested on 96 samples (74 non-cancer and 22 cancer). When combined with the Episwitch test, the binary PSA NPV decreased to 0.78 (95% CI 71.6% to 83.9%), while PPV increased to 0.81 (95% CI 62.4% to 92.4%) (Table 3). The test was further trained using The SHapley Additive exPlanations (SHAP), a method based on cooperative game theory and used to increase transparency and interpretability of machine learning models. The SHAP values for the best model of Episwitch-binary PSA test are shown in Figure 3 where a dot represents the value of each marker in an individual sample in the overall test performance. 

We have then used the PSA value in the test as a continuous variable similar to the five epigenetic markers. Tests were trained on 46 samples and then tested on 101 samples following random allocation and balancing of classes. The new multivariate Prostate Screening Episwitch test (PSE test see the Materials and Methods Statistical Analysis section) demonstrated a remarkable combined accuracy of 0.94 (95% CI 87.6% to 97.8%) with a PPV of 0.92 (95% CI 76.0% to 98.0%) and a NPV of 0.94 (95% CI 87.5% to 97.6%) (Table 3 and Figure 3). The Two cohorts were analysed separately, and in the PROSTAGRAM cohort, the PSE test had a PPV of 0.66 (95% CI 30.2% to 90.3%) and a NPV of 0.95 (95% CI 89.6% to 97.2%) with combined accuracy of 0.92 (95% CI 84.2% to 97.2%). In the relatively small Imperial cohort (established high risk PCa and cancer-negative controls), the test achieved a remarkable 100% detection rate with PPV, NPV, and a combined accuracy of 1 (95% CI 84.6% to 100%) (Table 3). Of note, combining the Episwitch test with neither MRI, no USS have not increased their performance. 

## 4. Discussion

### 4.1. Clinical Importance

In this study, we identified and internally validated a new PCa blood test composed of PSA and five chromosome conformations, originally discovered in association with advanced/stage III PCa [22], allowing disease diagnosis with very high accuracy. Our data demonstrate the presence of stable chromatin loops in the loci encoding for of *DAPK1*, *HSD3B2*, *SRD5A3*, *MMP1*, and *miRNA98.* Together with PSA taken as a continuous variable this epigenetic test has yielded a combined PPV of 0.92 and a NPV of 0.94 for PCa diagnosis in a mixed cohort with cancer prevalence of 34%. This is comparable to a hospital referral population with suspected PCa. Screening cohorts have a much lower cancer prevalence of 3% to 4%, and test performance is likely to be similar to results from the PROSTAGRAM cohort alone with a PPV of 0.66 and a NPV of 0.95.

The European randomised study of screening for PCa has shown significant reduction in PCa mortality in men who underwent routine PSA screening [10,29]. This notion was, however, not supported by the results of prostate, lung, colorectal, and ovarian (PLCO) cancer screening trial in the USA [30]. In either case, a total population screening yields a high volume of false positive results (due to low PPV of PSA test) leading to many unnecessary MRI and biopsy referrals resulting in significant costs, morbidity, patient worries, and overdiagnosis of clinically insignificant disease. The same problem of low PPV and high volume of referrals is the scourge of other screening modalities such as MRI and ultrasound [11]. 

Here we describe a new five-set biomarker panel based on PSA and qPCR readouts which is cost-effective, scalable, and easily accessed in most diagnostic facilities. This test requires only a small amount of blood, which is simple to collect and provides clinicians with rapidly available clinical readouts within hours. These time and cost savings together with an informative diagnostic decision bears a significant potential to fill the gap in the current protocols for assertive diagnosis of PCa. 

In our study we have used two patient cohorts. One from a population screening PROSTAGRAM study, which involved asymptomatic men randomly invited for cancer screening. This group of patients represented the “early cancers” cohort. The second group was from our previously published Imperial study [22] which involved men with established PCa diagnosis and high-risk PCa. 

In this study we have employed extensive machine learning algorithms in the test establishment, notably the BARA procedure and the SHAP method. This allowed us to build up on the existing Episwitch PCa test by adding PSA and removing *ERG_21* marker. Not surprisingly, the analysis showed that using PSA at cut-off of 3 ng/mL (as in the PROSTAGRAM study [11]) is inferior to use of PSA as a continuous variable. Indeed, previous studies have suggested that PSA dichotomy may notreflect the true nature of PCa [31].

In the isolated PROSTAGRAM cohort of 109 men, PSA at cut-off of 3 ng/mL had a PPV of 0.14 and a NPV of 0.93 with a combined accuracy of 0.79 (Table 3). The original PCa Episwitch markers for stage III PCa had a PPV of 0.8 and a NPV of 0.8 [22]. In the isolated PROSTAGRAM cohort of 109 men, the EpiSwitch PCa test alone has a PPV of 0.91 and a NPV of 0.32 with combined accuracy of 0.64 (Table 3). The new combined PSA–Episwitch (PSE) test demonstrated a combined accuracy of 0.94 with a PPV of 0.92 and a NPV of 0.94 (Table 3 and Figure 3). When two cohorts were analysed separately, the PSE test for the “early cancer” (PROSTAGRAM) cohort showed a PPV of 0.66 and a NPV of 0.95 with a combined accuracy of 0.92. In the established high risk PCa group (Imperial cohort) the PSE test achieved a 100% detection rate with PPV, NPV, and combined accuracy of 1 (Table 3). We surmise that much of the difference in results between the cohorts is due to different prevalence and different cancer types. 

### 4.2. Future Implementation of the PSE Test

Our data show that the PSE test is superior to PSA in PCa detection. The most significant difference is in PPV that increased from 0.14 for PSA alone to 0.66 for PSE in the screening cohort and 0.92 for PSE in the combined screening and advanced cancer cohort. It is worth noting that the screening cohort included insignificant cancers, an issue that should be addressed specifically as their detection does not provide significant clinical benefit, but rather distress to patients. The PSE test can be potentially utilized for both diagnostic and screening purposes, but each modality needs to be assessed in a separate trial as patient composition cohorts would be significantly different. Initially a prospective larger scale trial for the PSE test in a general population cohort with low cancer prevalence for PCa screening would be an immediate next step in confirming and expanding PSE test utility. The fact that the Episwitch test does not enhance diagnostic performance of MRI or USS suggests its suitability for determining whether these modalities are required as a follow-up to confirm diagnosis.

### 4.3. Pathophysiological Relevance of the Assay

Epigenetic analysis of CCs is a “holy grail” of cancer testing. The binary nature of the test (the chromosomal loop is either present or not) and the enormous combinatorial power (>10^10^ combinations are possible with ~50,000 loops screened) allow creating signatures that accurately fit clinically well-defined criteria. In PCa that would be discerning low-risk vs high-risk disease or identifying small but aggressive tumours and determining most appropriate therapeutic options. In addition, epigenetic changes are known to manifest early in tumorigenesis, making them useful for both diagnosis and prognosis [32]. This has transpired in our study which showed that epigenetic markers that were initially discovered in high-risk disease [22] are present in “early cancer” patients from the screening cohort [11]. This is consistent with early observations that regulatory changes in genome architecture and its remodelling could be very early pre-symptomatic events with the time lag towards build-up of the resulting pathological phenotype [33]. 

In this study we investigated regulatory genome architecture, also referred to as 3D genomics. This mechanism of regulation for cellular phenotypes operates at the interface of both genetic and epigenetic control mechanisms and is recognized today as an integrator of molecular events that contribute to the phenotype and clinical outcomes [34]. Consistent with this notion, the current study demonstrated that 3D genome architecture markers had synergy with the protein PSA biomarker when combined in a multivariant classifying model. This is not the only example of higher performance for an established biomarker modality in combination with EpiSwitch. Genomic risks, manifested in the form of single nucleotide polymorphisms (SNPs), have been long suspected to be associated with the anchoring sites of long-range interactions, thus affecting stability and off-rate for profiles of genome architectures [35]. 

As part of the background analysis for this study, early screening and evaluation of the developed EpiSwitch biomarkers associated with PCa [22,23], identified extensive overlap of 150 prostate SNPs from genome-wide association study (GWAS) Catalogue with anchoring sites for the prostate-associated EpiSwitch marker leads. This includes the Chromosome 8q24 region (Figure 4), all heavily associated with cancer (over 108 citations), with the MYC locus, as well as with several long non-coding RNAs loci—PCa Associated Transcript 1 (PCAT1) and PCAT2. Rendering of the individual genetic variant profiles into highly actionable data sets when matched with the 3D genomic EpiSwitch profiling will be a subject of a separate study. At this point, it is important to emphasize that the muti-variant combination of PSA with orthogonal data sets of 3D genomic biomarkers, which were already developed for stratification of PCa [22], significantly improved performance of non-invasive screening and stratification of patients at risk, for an early diagnosis of PCa. Three dimensional genomic profiling helps to disseminate informative components of other molecular readouts, with a closer link to clinical outcomes [34]. Of note SNPs were obtained from GWAS data [22,23]. No SNP analysis was possible in this study due to limitations in consent given for the original PROSTAGRAM study. 

Our data demonstrate the presence of stable chromatin loops in the loci encoding for *DAPK1*, *HSD3B2*, *SRD5A3*, *MMP1*, and *miRNA98* in the circulation of PCa patients. We have previously described their implication in PCa pathology [22]. String analysis has shown that at the protein level of four out of five markers belong to the same network with a high confidence of interaction (Figure 5). Despite the identification of these epigenetic loci, until recently, the mechanism of cancer-related epigenetic changes in PBMCs remained unidentified. We have previously identified that similar signatures existed in primary tumours [20,22]. Our recently published data show for the first time a proof of concept for horizontal transfer of chromosome conformations in cancer cell-monocyte co-culture without direct cell-cell contact [23]. 

### 4.4. Limitations

The limitations of this study include small number of patients, unavailability of other clinical indices like PHI and 4K (which are not part of the standard of care in the UK), retrospective setup, no follow-up (due to double anonymity), and high cancer prevalence in the significant cancers cohort. The screening PROSTAGRAM cohort contained both significant (Gleason 7) and insignificant (Gleason 6) cancers that were considered as a positive diagnosis. This was a pilot study establishing a new panel test for PCa diagnosis. Larger prospective blinded cohort validation studies would be required to further validate this test in the context of PCa diagnosis and screening. 

## 5. Conclusions

Our results demonstrate a multivariate model of the Prostate Screening Test (PSE) consisting of the standard continuous PSA test readout with specific set of blood-based, established PCa EpiSwitch biomarkers from the regulatory genome architecture of chromosome conformations (Episwitch). When tested in the context of screening population at risk, PSE yields a rapid and minimally invasive PCa diagnosis with a PPV of 0.92 and a NPV of 0.94. Due to its high PPV that significantly exceeds current screening modalities (due to its non-invasive nature and low costs), the PSE test can be utilized for both diagnostic and screening purposes, minimizing unnecessary referrals for expensive and invasive MRI and/or biopsy testing. Further prospective larger scale studies of the new PSE test in a population screening cohort with low cancer prevalence would be an immediate next step in confirming and expanding PSE test utility.

## Figures and Tables

**Figure 1 cancers-15-00821-f001:**
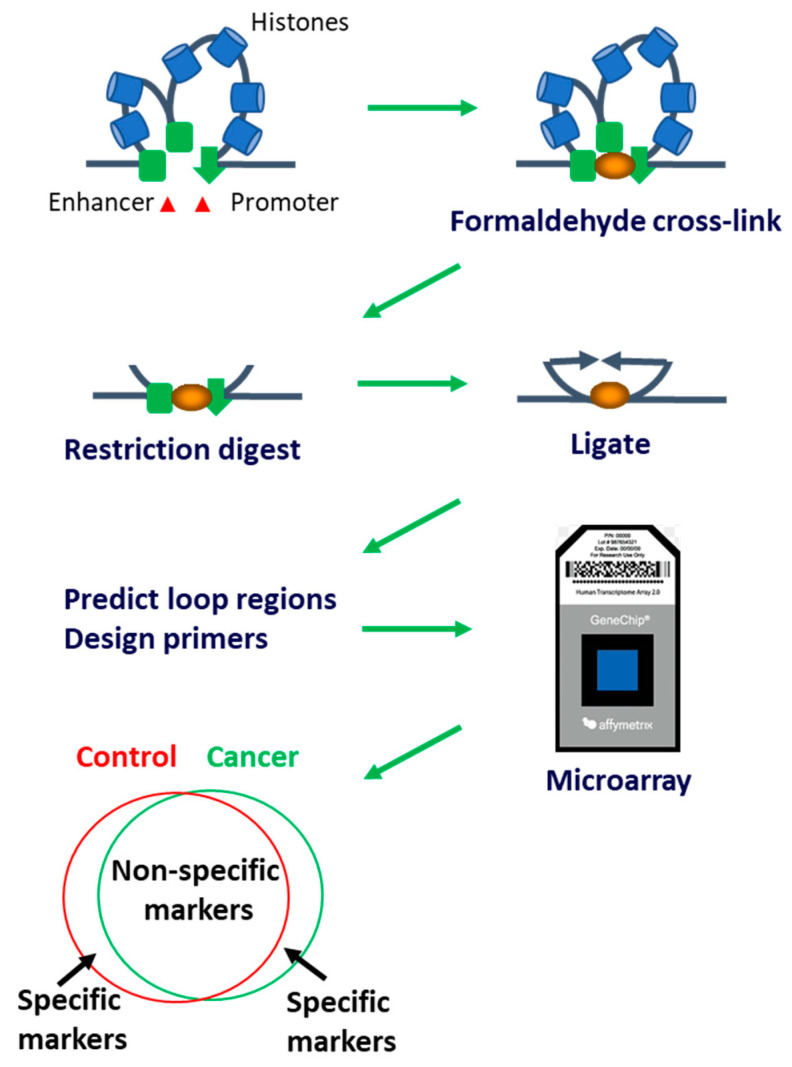
Scheme of chromatin conformation assay and microarray analysis. A three-dimensional structure of chromosomes contains loops with an enhancer and promoter regions, whereas the enhancer increases target gene promoter activity. CC capture assay: DNA is cross-linked using formaldehyde, digested, and ligated with the preference of cross-linked fragments. New sequences are formed where the loops have been. These new sequences are predicted via relevance machine vector algorithm. Specific primers to these sequences are synthesised and placed on the DNA microarray, which detects whether the loop was present or not. Resulting markers are analysed using multivariate analysis yielding specific epigenetic signatures for selected patient cohorts.

**Figure 2 cancers-15-00821-f002:**
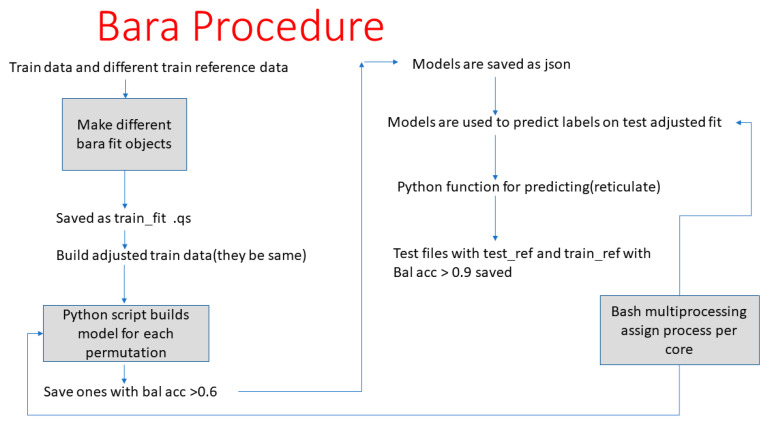
Batch adjustment by reference alignment (BARA) procedure. The scheme for batch adjustment by reference alignment (BARA) procedure [27] that was implemented in results analysis for better control of batch effects (non-biological variation).

**Figure 3 cancers-15-00821-f003:**
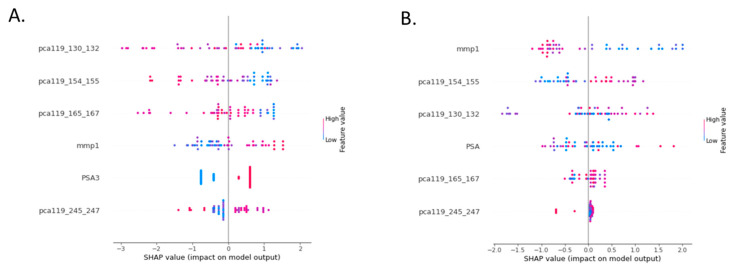
SHapley Additive exPlanations (SHAP) values of each marker contribution in the PSE model. SHAP values (26) demonstrating the contribution of each marker in the model are shown for the binary PSA (**A**) and continuous PSA (**B**). Each dot represents one sample.

**Figure 4 cancers-15-00821-f004:**
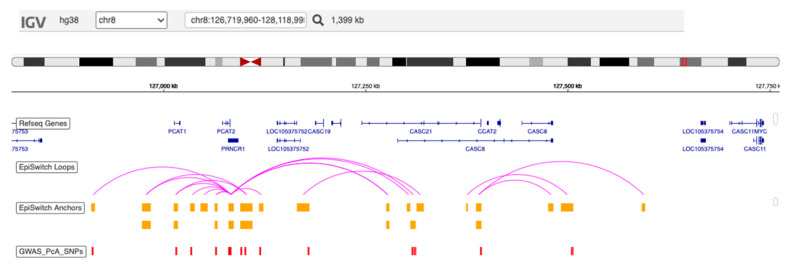
Schematic depiction of the overlap between the anchors of Episwitch loops and known PCa related SNPs on chromosome 8 from GWAS. The image shows the overlap of the anchoring sites of EpiSwitch 3D genomic markers, identified in early whole genome screening stage of EpiSwitch biomarker development [21,22], with positions of PCa SNPs from GWAS Catalogue on 8q24 chromosome region.

**Figure 5 cancers-15-00821-f005:**
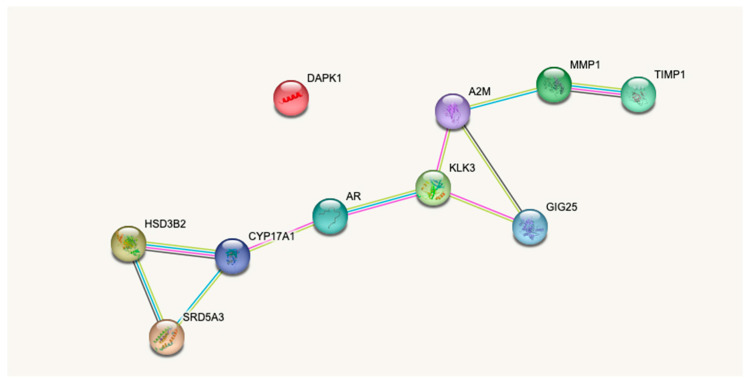
Schematic depiction of the genes involved in the PSE signature. The network was built using the five model markers and PSA, using the STRING DB (https://string-db.org/, accessed between 1 July and 1 October 2022). The network was generated by five additional entities that have a high confidence in interaction (data supporting the connections) suggesting high connectability at the protein level. *DAPK1*, *HSD3B2*, *SRD5A3* and *MMP1* model markers are circled in blue, *PSA (KLK3)* in red. miRNA98 is not in STRING.

**Table 1 cancers-15-00821-t001:** Individual accuracy of PSA, MRI, and ultrasound techniques in PCa detection in the PROSTAGRAM screening study [11]. Both significant (Gleason 7) and insignificant (Gleason 6) cancers that were considered as a positive diagnosis in this study.

PSA				
			Accuracy	0.87
	Cancer present	Cancer absent	Sensitivity	0.35
Test positive	13	27	Specificity	0.93
Test negative	24	342	PPV	0.32
			NPV	0.93
MRI				
			Accuracy	0.83
	Cancer present	Cancer absent	Sensitivity	0.57
Test positive	21	51	Specificity	0.86
Test negative	16	318	PPV	0.29
			NPV	0.95
Ultrasound				
			Accuracy	0.78
	Cancer present	Cancer absent	Sensitivity	0.59
Test positive	22	74	Specificity	0.79
Test negative	15	295	PPV	0.23
			NPV	0.95

**Table 2 cancers-15-00821-t002:** Number of patients in each cohort according to their clinical characteristics.

PROSTAGRAM Cohort	109
Control	88
Insignificant cancer	8
Significant cancer	13
PSA < 3	86
PSA > 3	23
Gleason sum = 0	88
Gleason sum = 6	13
Gleason sum = 7	8
Imperial Cohort	38
Control	9
Organ confined	9
Locally advanced	12
Metastatic	8
PSA < 3	4
PSA > 3	34
Gleason sum = 0	10
Gleason sum = 6	10
Gleason sum = 7	8
Gleason sum = 8	6
Gleason sum = 9	4

**Table 3 cancers-15-00821-t003:** Accuracy of various test modalities across both Imperial and PROSTAGRAM cohorts. Only final validation test (not training) results shown.

Binary PSA alone (across both cohorts)				
			Accuracy	0.79
	Cancer present	Cancer absent	Sensitivity	0.33
Test positive	2	12	Specificity	0.83
Test negative	4	61	PPV	0.14
			NPV	0.93
Continuous PSA alone (across both cohorts)				
			Accuracy	0.43
	Cancer present	Cancer absent	Sensitivity	0.66
Test positive	4	43	Specificity	0.41
Test negative	2	30	PPV	0.08
			NPV	0.93
Episwitch alone (across both cohorts)				
			Accuracy	0.64
	Cancer present	Cancer absent	Sensitivity	0.61
Test positive	16	34	Specificity	0.76
Test negative	5	54	PPV	0.91
			NPV	0.32
PSE with binary PSA (across both cohorts)				
			Accuracy	0.79
	Cancer present	Cancer absent	Sensitivity	0.53
Test positive	18	4	Specificity	0.93
Test negative	16	58	PPV	0.81
			NPV	0.78
PSE with continuous PSA (across both cohorts)				
			Accuracy	0.94
	Cancer present	Cancer absent	Sensitivity	0.86
Test positive	25	2	Specificity	0.97
Test negative	4	70	PPV	0.93
			NPV	0.95
PSE with continuous PSA (Prostagram cohort)				
			Accuracy	0.92
	Cancer present	Cancer absent	Sensitivity	0.5
Test positive	4	2	Specificity	0.97
Test negative	4	69	PPV	0.67
			NPV	0.95
PSE with continuous PSA (Imperial cohort)				
			Accuracy	1
	Cancer present	Cancer absent	Sensitivity	1
Test positive	21	0	Specificity	1
Test negative	0	1	PPV	1
			NPV	1

## Data Availability

The datasets used and/or analysed during the current study are available from the corresponding author on reasonable request.

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
