# Peer review of "Circulating Chromosome Conformation Signatures Significantly Enhance PSA Positive Predicting Value and Overall Accuracy for Prostate Cancer Detection"

_cancers, 2023, doi:10.3390/cancers15030821_

Round 1

Reviewer 1 Report

This study aims to determine whether combining Episwitch PCa test with PSA test will increase its diagnostic accuracy. The authors describe the utility of recently discovered epigenetic markers in the blood of PCa patients in PCa diagnosis and screening. The results demonstrate the additive utility of PSA and circulating chromosome conformations as biomarkers for PCA.

The article is well written. The results are well based on the comprehensive genetic and statistical methods used. The discussion is comprehensive, and conclusions match the results. The limitations acknowledgement is well outlined and is important for this kind of study.

Major comments – none

Minor comments:

  • Background: “Interestingly each screening modality had high NPV values of ~0.95” should read “Interestingly each screening modality had high NPV values of 0.93-0.95”
  • “and low PPV values of 0.22-0.32” should read “and low PPV values of 0.23-0.32”
  • Please, define the  purpose of your study in the abstract
  • Table 1: in prostagram study, were both significant and insignificant cancer counted in?
  • Why chromatin conformations are abbreviated CCCs, should it be chromatin conformation capture?
  • Prostagram in figure legends and tables should be in capital

Author Response

Response rev 1

Thank you very much for your assessment and comments.

In response to your specific comments and corrections:

  • Background: “Interestingly each screening modality had high NPV values of ~0.95” should read “Interestingly each screening modality had high NPV values of 0.93-0.95”

Amended as requested

  • “and low PPV values of 0.22-0.32” should read “and low PPV values of 0.23-0.32”

Amended as requested

  • Please, define the  purpose of your study in the abstract

Added as requested

  • Table 1: in prostagram study, were both significant and insignificant cancer counted in?

Yes, as this study was a pilot – both types of prostate cancer were counted in. This is data from the previous study (ref 11). We have reflected on this in the limitations part and have added this info to the table legend as suggested. We have added a future implementation part where we have discussed this in more detail.

  • Why chromatin conformations are abbreviated CCCs, should it be chromatin conformation capture?

Abbreviations corrected as requested (changed to CC).

  • Prostagram in figure legends and tables should be in capital

Amended as requested

Reviewer 2 Report

In this paper entitled “Circulating chromosome conformation signatures significantly enhance PSA positive predicting value and overall accuracy for prostate cancer detection.” aims to determine whether combining Episwitch PCa test with PSA test will increase its diagnostic accuracy. The authors have used available full blood samples from patients enrolled into a prospective prostate cancer screening pilot study (PROSTAGRAM) well as samples of patients with either an established PCa diagnosis or confirmed cancer-negative controls from Imperial College NHS Trust. The results demonstrated a multivariate model of Prostate Screening Test (PSE) consisting of the standard continuous PSA test readout with specific set of blood-based, established PCa EpiSwitch biomarkers from the regulatory genome architecture of chromosome conformations. The paper is well written and has the makings of a publication. Can authors could provide as guidelines for clinical implementation and how to implement?

Author Response

Response rev 2

Thank you very much for your assessment and comments.

Can authors could provide as guidelines for clinical implementation and how to implement?

We have added a paragraph regarding future implementation of the PSE test.

Reviewer 3 Report

This is overall well written paper describing the new combined test for prostate cancer combining commonly used PSA and new epigenetic biomarkers.

Authors use chromosome conformations as potential markers for prostate cancer occurrence. Interestingly the chromosomal conformations are assessed in circulating blood cells and not in cancer cells. This is based on the previous studies of the authors showing that cancer cells exchange epigenetic information with circulating cells, which can be measured. Authors use a previously determined set of epigenetic signatures in two cohorts of patients – one with low and the other with high cancer prevalence. Combining this set with PSA data offers a remarkable improvement for the PSA performance for prostate cancer diagnosis.

This paper is on an intersection between epigenetics, clinical urology, oncology and biomarker statistical analysis. This new study is based on scientifically proven methods and hypothesis, use complex statistical methods, and perhaps is mainly of interest to clinicians working on prostate cancer as the findings have significant clinical relevance.

I have following minor comments/corrections:

Authors use abbreviation (PCa) for prostate cancer, but then use words again multiple times.

Did epigenetic test provide any diagnostic improvement to MRI or USS in the prostagram cohort?

In background section the “magnetic resonance imaging (MRI) is used twice on first page. Please correct.

On page 3 it must be written “in primary prostate tumours (18), not “Un primary prostate tumours”.

In the “BARA procedure” the blue colour in the squares is too dark. Can you use another colour (e.g. light grey)?

Prostate cancer revealed SNPs are mentioned in the discussion, figure 4. Have any SNPs been analysed on the patients participating in this study?

Author Response

Response to reviewer 3.

Thank you very much for your assessment and comments.

Authors use abbreviation (PCa) for prostate cancer, but then use words again multiple times.

Amended as requested

Did epigenetic test provide any diagnostic improvement to MRI or USS in the prostagram cohort?

Unfortunately, it did not. Therefore suitable only for pre-screening or pre-diagnosis – determining whether these modalities are required as a f/up to confirm diagnosis. A paragraph was added to manuscript.

In background section the “magnetic resonance imaging (MRI) is used twice on first page. Please correct.

Amended as requested

On page 3 it must be written “in primary prostate tumours (18), not “Un primary prostate tumours”.

Amended as requested

In the “BARA procedure” the blue colour in the squares is too dark. Can you use another colour (e.g. light grey)?

Amended as requested

Prostate cancer revealed SNPs are mentioned in the discussion, figure 4. Have any SNPs been analysed on the patients participating in this study?

No genetic analysis was done in this study due to limitations in consent given for the original PROSTAGRAM study. SNPs were used from the literature search. The paper was amended to reflect that specifically.
